# Effect of Pre-Emulsion of Pea-Grass Carp Co-Precipitation Dual Protein on the Gel Quality of Fish Sausage

**DOI:** 10.3390/foods11203192

**Published:** 2022-10-13

**Authors:** Xiaohu Zhou, Chaohua Zhang, Liangzhong Zhao, Xiaojie Zhou, Wenhong Cao, Chunxia Zhou

**Affiliations:** 1College of Food Science and Technology, Guangdong Ocean University, Zhanjiang 524088, China; 2College of Food and Chemical Engineering, Shaoyang University, Shaoyang 422000, China; 3Guangdong Provincial Key Laboratory of Aquatic Products Processing and Safety, Zhanjiang 524088, China; 4Hunan Provincial Key Laboratory of Soybean Products Processing and Safety Control, Shaoyang 422000, China; 5Collaborative Innovation Center of Seafood Deep Processing, Dalian Polytechnic University, Dalian 116034, China

**Keywords:** pea, dual protein, plant protein, co-precipitation dual protein, blended dual protein, fish sausage, pre-emulsion

## Abstract

Currently, the processing method of introducing plant protein into meat products has attracted great attention. However, the direct addition of plant protein often leads to a decline in meat product quality. This paper aims to provide an efficient method for incorporating plant protein into fish sausage. Pea protein isolate (PPI), grass carp protein isolate (CPI) and pea-grass carp coprecipitated dual protein (Co) were derived from pea and grass carp by an isoelectric solubilisation/precipitation method. At the same time, the blended dual protein (BL) was obtained by blending PPI with CPI, and the plant and animal protein content of Co and BL was both controlled to be the same. The four proteins were combined with soybean oil and water to form a three-phase pre-emulsification system of protein-oil-water, which was added to grass carp meat as a replacement for animal fat to prepare fish sausage. The gelation properties of the four fish sausages and those without protein were analysed. The results showed that the gel quality of PPI fish sausage is poor, while the overall quality of Co fish sausage as a whole was significantly superior to that of PPI and BL, which was equivalent to CPI fish sausage. The sensory score of the Co fish sausage was slightly lower than that of CPI, but it had significantly higher water-holding capacity and hardness (*p* < 0.05). The Co fish sausage showed the synergistic effect of heterologous proteins, while BL had some antagonistic effects. This study shows that Co pre-emulsion is an effective strategy to introduce plant protein, so it has a good application prospect in the meat industry.

## 1. Introduction

Fish represent a high-quality animal protein with high protein content and bioavailability [1]. China has abundant fish resources, with the total production of fish resources reaching 35.21 million tonnes in 2020 [2]. Making surimi products from fish, such as fish sausage, is popular among consumers because of its distinct flavour and low price. However, traditional fish sausages have a high fat content, which has some defects, such as a rough gel structure and health risks [3]. At present, there is a processing trend to use plant-based components instead of animal components [4,5].

Pre-emulsification is the emulsification of oil with food protein itself as a natural emulsifier, which can form a crosslink with continuous-phase protein gel matrix by covalent or non-covalent bond [6]. Liu [7] found that soy protein pre-emulsified plant oil significantly increased the whiteness and water-holding capacity of surimi. However, the above study showed that when the proportion of plant protein increased, the quality deteriorated obviously. With the rise of plant-based meat, pea protein is favoured by researchers. A series of studies on pea protein added to pork [8], beef [9], and chicken [10] gels have been conducted. It was demonstrated that pea protein could increase the thermal denaturation temperature of myosin head structure, thus improving gel-forming ability and gel quality significantly. Due to the limitation of using a single protein, practical processing is more likely to use two or more proteins. Recent research shows that the addition of dual proteins can significantly improve the hardness, springiness, cooking loss, and water-holding capacity of surimi gel, especially when the ratio of plant and animal protein is 1:1 [11]. It can be seen that dual protein is a feasible way to introduce plant protein into meat products. However, there have been few reports on the application of pea protein and its compound dual protein in surimi.

Dual proteins can be divided into blended dual protein (BL) and coprecipitated dual protein (Co) according to their preparation methods [12]. BL refers to the protein produced by the direct mixing of protein isolate. Studies have shown that BL from peas and cods has superior emulsification properties than a single protein, reflecting the synergistic effect [13]. However, studies have confirmed that as the content of plant protein increases due to different species of BL proteins, the incompatibility of the molecular configurations of heterologous proteins causes steric hindrance and damage to gel structures (such as globulin and myofibril), resulting in antagonistic effects on gel properties [14,15]. Co was prepared using the isoelectric solubilisation/precipitation (ISP) method. Driven by pH, heterogeneous proteins are dissolved and precipitated in the same dispersion system to promote interactions between heterogeneous proteins, such as the formation of disulphide bonds and changes in protein subunit composition, surface charge, solubility, and surface hydrophobicity, thus effectively improving functional properties [16]. In our previous study, it was proved that Co has superior emulsification [17] and gel [18] properties than BL and a single protein. Additionally, we found that pre-emulsifying soybean oil can improve the gel quality of fish sausage [19]. However, no literature has reported the actual performance of the emulsification and gel properties of BL and Co in fish sausage.

In this study, a “protein-oil-water” three-phase pre-emulsion system was constructed, and pre-emulsion of PPI, CPI, and BL and without protein were used as controls to explore the effect of Co on the characteristics (such as water-holding capacity and TPA) of fish-sausage gel. The aim of this study was to find a more ideal way to introduce plant protein into fish sausage than the direct addition method so as to improve the gel quality of fish sausage and expand the application scope of Co in food processing. This study can provide a theoretical basis for increasing the application of pea protein in animal-based food.

## 2. Materials and Methods

### 2.1. Materials

Split peas were purchased from Foshan Jinnuoyi Processing Factory of Agricultural Products (Foshan, China); the protein content was 21.12% (6.25 of N, wet weight). The grass carp was purchased from the local fish market in Shaoyang City, Hunan Province. The gills, heads, tails, bones, and blood were all removed, leaving only the white muscles on the back, which were retained and frozen at a low temperature. The fish muscles were brought back to the laboratory and stored at −20 °C; the protein content was 18.27% (6.25 of N, wet weight). Natural-flavoured rice wine was purchased from Shandong Luhua Group Co., Ltd. (Yantai, China). Edible salt was purchased from China National Salt Corporation (Beijing, China). Edible-grade sodium nitrite and edible-grade sodium tripolyphosphate were purchased from Sichuan Jinshan Pharmaceutical Co., Ltd. (Meishan, China) and Hubei Xingfa Co., Ltd. (Yichang, China), respectively. Five-spice powder and pepper powder were purchased from the supermarket in Shaoyang City, Hunan Province. Sheep casings were purchased from the Xuchang Shenyuan casing company (Xuchang, China). Other analytical grade reagents were ordered from the Guangzhou Chemical Reagent Factory (Guangzhou, China).

### 2.2. Preparation of PPI, CPI, Co, and BL

PPI, CPI, and Co are prepared using the ISP method as described in reference [17], where Co is prepared by dissolving and precipitating pea powder and fish muscle in the same container simultaneously. Peas were powdered and sieved through a 100-mesh sieve (0.15 mm). The pea powder was mixed with deionised water at 4 °C in a ratio of 1:9 (*w*/*v*). The pH was adjusted to 10.0 by dissolving 1 M NaOH for 30 min, followed by centrifugation at 10,000× *g* and 4 °C for 20 min. The precipitate was removed, and 1 M HCl was slowly added to the supernatant. The mixture was fully stirred, and the pH was adjusted to 5.0. After washing the precipitation with deionised water and adjusting the pH to 7.0, dialysis was performed. PPI was obtained after 48 h of freeze drying. The fish muscles were defrosted and chopped, and CPI was prepared according to the steps indicated above for PPI. The pea powder was mixed with fish muscle, and Co was prepared following the steps described above. BL was obtained by directly mixing powdered PPI and CPI. To ensure a similar comparison, the protein ratio of pea and grass carp in BL and Co was kept constant at 1:1 (W:W) in this study.

### 2.3. Non-Reduced SDS-PAGE

The original protein compositions of BL and Co were characterised by non-reduced sodium dodecyl sulphate-polyacrylamide gel electrophoresis (non-reduced SDS-PAGE) as follows. Solutions of 1% protein concentration were prepared. The volume fraction of concentrated gel (upper gel) was 5%, while the volume fraction of separated gel (lower gel) was 12%. No reducing agent dithiothreitol (DTT) was added; the protein marker was 16–270 kDa. Protein band abundance was expressed by relative optical density (%) measured using Image LabTM V4.0 software (Bio-Rad Laboratories, Hercules, CA, USA).

### 2.4. Determination of Chemical Composition of PPI, CPI, Co, and BL

Chemical compositions, including moisture (AOAC 935.29), ash (AOAC 938.08), protein (AOAC 2001.11), and fat (AOAC 920.39), were determined according to the methods recommended by AOAC [20]. Carbohydrates were calculated using the following formula: 100 − (moisture + ash + protein + fat).

### 2.5. Preparation of Pre-Emulsion

The protein powder of PPI, CPI, BL, and Co was used as an emulsifier to prepare pre-emulsified soybean oil [21]. The material was weighed using a mass ratio of m (protein powder):m (soybean oil):m (water) = 12:44:44. The protein was mixed with water, slowly stirred for two hours, and then mixed with soybean oil. The mixture was evenly dispersed with a T18 homogeneous shear machine (IKA Laboratory Equipment, Staufen, GER) at 11,000 rpm for one min in an ice bath, with a five-minute interval, and repeated five times and kept at 4 °C. At the same time, the blank control group without protein was established, and the homogenisation method was the same as that of M (protein powder):m (soybean oil):m (water) = 0:44:44.

### 2.6. Preparation of Fish Sausage

Formula: Fish sausage (g/100 g) was determined as 64.0 g fish meat, 30.0 g pre-emulsion, 3.0 g rice wine, 2.0 g edible salt, 0.5 g five-spice powder, 0.3 g tripolyphosphate, and 0.2 g pepper. In addition, 0.01 g sodium nitrite was added to prevent *Clostridium botulinum*. The fish-sausage emulsion in the blank control group was 26.4 g.

Marinating: Mince the fish into small pieces, add the marinade (soybean oil, edible salt, tripolyphosphate, five-spice powder, pepper, sodium nitrite) into it, and marinate it at 4 °C for 48 h.

Chopping: Mix the pickled lean meat with the pre-emulsion, then slowly chop and mix at a low temperature (10 °C) for five minutes.

Pouring: After eliminating the air pockets, use a small enema device to pour into the sausage casing, tie knots every 10 cm, count by weight, and control the diameter at about 2.5 cm.

Cooking: Fish sausage was placed in a 90 °C constant temperature water bath for 45 min, cooled in an ice-water bath for 15 min, drained of water on the surface of the casings, weighed and counted, and stored in 4 °C for 8 h.

### 2.7. Cooking Loss

The cooking loss (CL) of the gel was determined using the Youssef method [22]. The gel was cut into a cylinder of about 15 × 15 × 5 mm and weighed (W_1_), then put into a cooking bag and sealed. After cooking for 20 min in a 90 °C water bath, it was stored at 4 °C in a refrigerator for 24 h. Again, the gel was weighted (W_2_) after carefully blotting its surface with filter paper. The CL was calculated as follows:(1)$$\mathrm{CL}(\%)=\frac{W_{1}-W_{2}}{W_{1}}\times 100$$
where W_1_ is the mass before cooking (g), and W_2_ is the mass after cooking (g).

### 2.8. Water Holding Capacity

Next, we wrapped 8 g gel samples in double filter paper and placed them in a centrifugal tube before weighing the total weight. After centrifuging at 5000× *g* at 4 °C for 10 min, the gel was taken out and weighed. Formula 2 is used to calculate WHC, where M is the mass of centrifugal tube (g); M_1_ is the mass of gel and centrifugal tube before centrifugation (g); M_2_ is the mass of gel and centrifugal tube after centrifugation (g).
(2)$$\mathrm{WHC}(\%)=(M_{2}-M)/(M_{1}-M)\times 100$$

### 2.9. Folding Test

According to Kudo [23], the fish sausage was cut to a thickness of 3 mm and extruded between the thumb and index finger. The folding ability of the fish sausages was divided into five levels: (1) Sausage slices were broken into two pieces on the first folding; (2) sausage slices were cracked but not broken on the first folding; (3) sausage slices did not show any cracks on the first folding, but they were broken into two pieces on the second folding; (4) sausages slices did not show cracks on the first folding but did show cracks on the second folding; and (5) sausage slices did not show cracks even after the second folding.

### 2.10. TPA

Texture profile analysis were measured by a P35 cylindrical flat bottom probe with a texture analyser (LS5, Ametek, Berwyn, PA, USA). Parameter setting: sample surface flatness height of 20 mm; front, middle, and final test speed of 4.0 mm/s, 3.0 mm/s, and 4.0 mm/s; descent distance of 40%; compression time of 5 s; trigger force of 5 g.

### 2.11. Colour

The colourimeter was used to measure the colourimetry of fish sausage, which was represented by the L*, a*, and b* values. Whiteness is calculated as follows:(3)$$\mathrm{Whiteness}=100-\sqrt{(100-L*{)}^{2}+{a*}^{2}+b{*}^{2}}$$

### 2.12. LF-NMR

Low-field nuclear magnetic resonance (MQC-23, Oxford, UK) was used to determine the water distribution of the fish sausage [24]. The fish sausage was packed into a 20 cm NMR tube, sealed with plastic wrap, and the transverse relaxation time (T) of the protein gel was measured by Carr–Purcell–Meiboom–Gill (CPMG) pulse sequence. The parameters were set as: P90 = 6, P180 = 12, SFI = 23.4 MHz, DW = 1.0, SW = 1,000,000 Hz, SI = 1, NECH = 256, and NS = 16; WinDXP software was used to invert CPMG and save its data. The relaxation time diagram was made by drawing software. The peak area in the atlas was accumulated, and the peak area represented the percentage content of water in the group.

### 2.13. SEM

The samples were cut into 3 mm × 2 mm and placed in a covered container. They were soaked in 2.5% pH 7.2 phosphate buffer at 4 °C for more than eight hours and then washed three times with 0.1 mol/mL pH 7.2 phosphate buffer for 10 min each. Gradient elution was performed with 50%, 70%, 80%, and 90% ethanol solutions for 10 min each. Elution was repeated three times with 100% anhydrous ethanol for 10 min each. The dehydrated samples were defatted with the proper amount of trichloromethane for one hour, then replaced with a mixture of tert-butyl alcohol and ethanol with a volume ratio of 1:1 for 15 min, and after that replaced with 100% tert-butyl alcohol for 20 min and freeze-dried for 48 h. The surface morphology of the sample was observed by a scanning electron microscope (Gemini300, Zeiss, Oberkochen, Germany) under a 5000× field of vision.

### 2.14. Sensory Evaluation

The sensory evaluation team of sausages comprised ten teachers and postgraduates (five males and five females, aged 24–35) from the major of food Science and Engineering at Shaoyang University. All of these people have experience with food sensory evaluation and were familiar with the experimental process and operating norms. Prior to the analysis, all participants (volunteers) in this experiment were informed of the aim of the sensory evaluation, and they provided informed consent to perform it. The group members’ scores ranged from 0 (poor sensorial property) to 10 (excellent sensorial property), indicating that their expectations for appearance, smell, taste, texture, and overall acceptability ranged from very low to very high. The scoring criteria are shown in Table 1. Sausages were re-heated in water for two minutes and kept in an oven at 40 °C before being served to the panel randomly on white porcelain plates under natural light at room temperature. A glass of water was also provided to cleanse the palate.

### 2.15. Statistical Analysis

SPSS 25.0 software of IBM company was used for statistical analysis. The statistical differences between the two groups were analysed by one-way analysis of variance (ANOVA) and Duncan’s multiple range test (*p* < 0.05). The original Pro 2021 software was used for drawing.

## 3. Results and Discussion

### 3.1. Protein Composition Analysis

Non-reducing electrophoresis cannot disrupt the disulphide bonds. Therefore, it reflects the original composition of the various proteins in a mixed protein (Figure 1 and Table 2), and both BL and Co contained myosin heavy chain (MHC, ~200 kDa), convicilin (~70 kDa), legumin (~62 kDa), vicilin (~50 kDa), actins (AC, ~42 kDa), legumin α (~38 kDa), tropomyosin (TM, ~34 kDa), and legumin β (~22 kDa), all aggregates consistent with the literature [25,26]. Compared with legumin α + β, vicilin is more flexible and exhibits higher interfacial activity [27]. Many studies have confirmed that the vicilin/legumin a + B ratio is positively related to a protein’s functional properties, including solubility and foaming and emulsifying properties [28,29]. Thus, the ratio of vicilin to legumin a + β is very important. As shown in Table 2, the proportion of Co (203.99%) is 2.82-fold larger than that of BL (72.36%). Therefore, the analysis of protein composition indicates that the functional properties of Co might be superior to those of BL.

### 3.2. Chemical Composition of PPI, CPI, Co and BL

The chemical composition analysis of PPI, CPI, BL, and Co is shown in Table 3. The protein content of the four protein samples was about 85%, which belongs to high-quality protein powder, indicating that the ISP method is an effective way to concentrate pea protein and grass carp protein. Co was the highest (*p* < 0.05) at 87.91%, indicating that Co has the potential to be used as the raw material of high-protein food. Compared with BL, Co contains less fat. When Co is added to fish sausage, the damage to the gel structure and nutritional value caused by fat oxidation can be reduced, and the shelf life of the product can be prolonged [30]. We previously speculated that the carbohydrate content of Co should be between PPI and CPI. Interestingly, experimental data showed that the carbohydrate content of Co was only 0.16% lower than CPI (*p* < 0.05), suggesting that co-precipitation enhanced the removal rate of starch or muscle glycogen. It is speculated that this is because the interaction of heterologous proteins reduces protein–carbohydrate (glycoprotein) binding during co-dissolution or precipitation. In addition, Co has low fat and low carbohydrate characteristics, which is in line with the consumption trend. Therefore, Co, as an isolated protein powder, has greater application potential in meat products than BL.

### 3.3. Cooking Loss

The effect of adding different protein pre-emulsified soybean oil on the CL of fish sausage is shown in Figure 2. In this study, due to different emulsifiers, the introduction of soybean oil in the form of protein pre-emulsification will affect CL. CPI fish sausage had the lowest cooking loss (5.24%, *p* < 0.05), indicating that adding fish protein homologous to the fish-sausage matrix in the pre-emulsification emulsion can reduce cooking loss [31]. Compared with the control group, the cooking loss of PPI fish sausage was higher (*p* < 0.05). This is similar to the finding that replacing pig fat with pre-emulsified olive oil stabilised by soy protein isolate increased cooking loss. This is because the plant protein in fish sausage is not tightly bound to the fish matrix, resulting in a loose structure of the sausage that makes it difficult to hold water [32]. However, there was no significant difference in the CL of BL and Co when compared with the control group (*p* > 0.05). It can be seen that the introduction of plant protein in the form of dual proteins in fish sausage can significantly reduce CL.

### 3.4. Water-Holding Capacity

The WHC can reflect the cross-linking degree of the blend gel of pre-emulsified protein and fish meat. The effect of soybean oil pre-emulsified with four kinds of proteins on the water-holding capacity of fish sausage is shown in Figure 3. The results showed that the WHC of the five kinds of fish sausage was greater than 93%, with the control group having the lowest (93.20% *p* < 0.05), indicating that increased protein concentration results in a stronger gel network. This is consistent with the results reported in the literature [33]. The WHC of Co-, PPI-, and CPI fish sausage was higher than 96.00%. Under centrifugation, the water in the capillary of the gel permeates the surface and is removed by filter paper. Therefore, it can be considered that the addition of Co, PPI, and CPI reduces the flow of free water [34]. In addition, the WHC of Co fish sausage was the highest, while that of BL fish sausage was the lowest. Combined with the previous results [18], we speculate that this is due to the antagonistic effect of PPI and CPI in BL on heat-induced gelation, which leads to a loose structure and poor WHC of fish sausage. It can be seen that Co is more beneficial to the water retention of fish-sausage gels than BL.

### 3.5. Folding Test

The folding test can directly reflect the gel strength of fish sausage and allow researchers to quickly understand its texture characteristics [35]. The folding test results of the five kinds of fish sausages are shown in Figure 4, which confirms the results of CL and WHC. The folding ability of the control group was lower than that of PPI-, CPI-, and Co fish sausage, which was attributed to the weak gel network formed by the lower protein content [36]. However, the lowest folding ability of BL fish sausage was only level 3 (*p* < 0.05). This is due to the antagonism of PPI and CPI in BL, leading to the gel structure loosening.

### 3.6. TPA

The TPA results of the five sausage samples are shown in Table 4. The hardness, cohesiveness, and chewiness of fish sausage added with Co were significantly higher than those of PPI and CPI (*p* < 0.05), which indicated the synergistic effect of Co. According to previous studies [18], it is speculated that disulphide bonds account for the highest proportion of intermolecular forces of Co gel, which makes the hardness and cohesion of Co fish sausage stronger and thus necessitating greater energy consumption during chewing. However, the thermal denaturation temperature of PPI (82 °C) [37] and CPI (56 °C) [38] contained in BL is obviously different, which leads to the loose structure of heat-induced gel formed by BL, which leads to the lowest hardness, cohesion, and chewiness of BL fish sausage (*p* < 0.05), showing an obvious antagonistic effect. The hardness, cohesion, and chewiness of the blank control group were the lowest, but the springiness was the highest (*p* < 0.05). This is similar to the research results of improving the texture characteristics of fish sausage by adding pea protein [39]. Adding protein can enhance the interaction between protein molecules in surimi gel, thus improving the texture characteristics of the gel.

### 3.7. Colour

Colour impacts its application in food, and colour tests can objectively evaluate the colour of protein samples. L* represents sample brightness; the positive value of a* denotes redness, and the negative value represents greenness. A positive value of b* is yellowness, and a negative value is blueness. The effect of different protein pre-emulsified soybean oil on the colour of fish sausage is shown in Figure 5 and Table 5. The brightness (L*) and whiteness of PPI fish sausage were lower than those of the other groups (*p* < 0.05), and yellowness (b*) was higher than those in other groups (*p* < 0.05). This is related to the yellowish colour of PPI, which is attributed to the natural yellow plant pigments in peas that are thermally stable. The brightness (L*) of CPI fish sausage was the highest, reaching 86.37 (*p* < 0.05), and the yellowness (b*) was the lowest (*p* < 0.05), but the whiteness was significantly higher than that of other groups (*p* < 0.05) [40]. The redness/greenness (a*) of the fish sausage was lower in the five groups, which had little influence on the colour. The four colour indices of the fish sausage in the control group, i.e., BL and Co, were all between PPI and CPI, indicating that the relative white of CPI had a masking effect on the relative yellow of PPI [41]. Therefore, adding plant protein in the form of dual proteins can reduce the adverse effect of yellow pigments in plants on fish sausage, and the colour quality of BL is superior than that of Co.

### 3.8. Water Distribution Analysis

Low-field nuclear magnetic resonance (LF-NMR) is a useful method to study the content, state, distribution, migration, diffusion, and the interaction of water molecules with other molecules. These properties of water are the key factors in determining the quality of fish-sausage gel [42]. Figure 6 shows the effect of soybean oil pre-emulsified with different proteins on water distribution in fish sausage. The relaxation time T of hydrogen protons in fish sausage reflects their degree of binding or freedom. The higher the T, the lower the binding force of hydrogen protons, leading to decreased stability and increased fluidity. The peak area of each stage represents water content with a different binding degree in fish sausage. T1 represents the bound water tightly adsorbed with macromolecules such as protein; T2 represents the water locked in the gel network of fish sausage, which has a certain fluidity and is classified as not easily allowing the flow of water; and T3 represents the water outside the gel network structure of fish sausage: free water with high fluidity [43]. It can be seen from Figure 6 that the T1 peak area of fish sausage containing pea protein components, such as PPI-, BL-, and Co fish sausage, is more significant. This may be due to the existence of pea protein in the gel in the form of filling after emulsification, and the degree of interaction with the gel matrix is weak. In contrast, the adsorption capacity of plant protein (PPI) to water is considerably stronger than that of fish protein, resulting in a higher volume of bound water (T1) [44]. Only the BL fish sausage had a pronounced T3 peak, indicating that it had a high free-flowing water content and a low gel water-holding capacity. However, the difference between T1 and T2 peaks of CPI fish sausage was not noticeable, and the peak distribution was similar to that of the control group, indicating that the CPI bonded well with the fish gel matrix during gelation, leaving appropriate gaps to lock the adsorbed water in the gel network [45]. Generally speaking, the water distribution of Co fish sausage was between that of CPI- and PPI fish sausage, with no discernible T3 peak. Again, BL fish sausage showed an antagonistic effect.

### 3.9. Microstructure

To accurately understand the characteristics of the protein emulsion and fish matrix blend gel, the microstructure of the protein pre-emulsified fish-sausage gel was observed by scanning electron microscopy at 40,000×, as shown in Figure 7. Compared with the microstructure of protein gel in the previous report [18], the gel network structure of fish sausage in the five groups was more compact and even, which could be attributed to the high content of TG enzyme in fresh surimi and forms a higher proportion of covalent cross-linking [46]. The microstructure of the control group and CPI fish sausage was compact and consistent with the results reported in other literature, which indicated that CPI participated in the formation of fish-sausage gel through cross-linking [41]. However, CPI fish sausages may show to an uneven distribution of TGase [47].

The presence of pea globulin particles in PPI-, BL-, and Co sausages confirmed that plant protein components filled the gel network. One possible explanation is that CPI acts as filler for the holes in the fish gel structure, allowing the two separate structures to interact and become intertwined to form a homogeneous structure [39,48,49]. The PPI was aggregation-filled on the gel’s surface, which was different from the fish matrix gel. However, there are many large holes in BL fish sausage, which are consistent with the above research results and are attributed to the antagonistic effect of PPI and CPI. It is worth noting that Co fish sausage also has a few holes, but the overall structure of the gel is dense. Co is “embedded” in the gel matrix, which is beneficial to lock water and improve gel strength, confirming that the common fish sausage has the highest WHC and hardness.

### 3.10. Sensory Evaluation

Sausages were tested for sensory attributes, such as appearance, texture, taste, smell, and overall acceptability. It can be seen in Figure 8 that the control group had the lowest taste score, indicating that protein pre-emulsification improves the taste of fish sausage. CPI fish sausages had the highest scores except for texture (*p* < 0.05). The scores of appearance and overall acceptability were 7.22 and 7.56, respectively. This is due to the whiteness, smell, and taste of CPI, which are all similar to that of fish. Similar observations were made by Santana, who reported that the addition of fish protein into sausages did not cause changes in their smell, oiliness, and colour [50]. PPI fish sausage scored the lowest in appearance because of its yellow colour.

Texture was the main attribute that affected consumer preferences [51]. The texture score of Co fish sausage was higher than CPI (*p* < 0.05), and the overall acceptability was second only to CPI, indicating that the colour and smell of pea components would affect the sensory quality of the fish sausage. It is noteworthy that the overall acceptability, texture, and smell of Co fish sausages are significantly higher than those of PPI- and BL fish sausages. The results were consistent with the folding test level, WHC, and texture properties, which again indicated that Co addition was a more ideal method for introducing plant protein compared to PPI and BL.

## 4. Conclusions

On the basis of the previous research on the emulsion and gel properties of dual proteins (BL and Co), the quality of fish-sausage gel formed by pre-emulsification of PPI, CPI, BL, and Co was discussed in this paper. Comprehensive analysis showed that adding protein pre-emulsified soybean oil to fish sausage could effectively improve the gel network and edible quality of fish sausage. Because CPI and fish-sausage matrix gel are homologous, fish-sausage gel prepared using the CPI pre-emulsion method has a higher quality. The sensory score of Co-fish-sausage gel was slightly lower than that of CPI. However, because Co has the functional characteristics of both grass carp protein and pea protein, it has significant advantages in terms of WHC, hardness, and microstructure. This shows that the overall quality of Co-fish-sausage gel is comparable to or even slightly higher than CPI. In addition, the quality of PPI- and BL-fish-sausage gel is poor, with BL having a particularly pronounced antagonistic effect. Therefore, co-precipitation dual protein is an effective method for introducing plant protein into fish sausage. This study provides a theoretical reference for the application of Co in meat products. We will further explore the influence of the combination and ratio of other plant and animal proteins on the food gel system.

## Figures and Tables

**Figure 1 foods-11-03192-f001:**
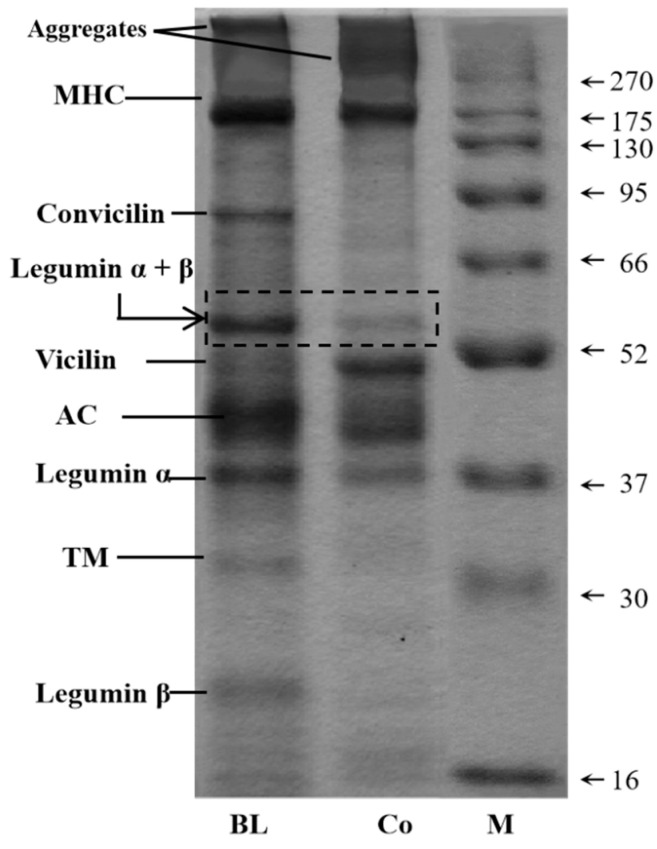
Non-reducing electrophoretograms of BL and Co.

**Figure 2 foods-11-03192-f002:**
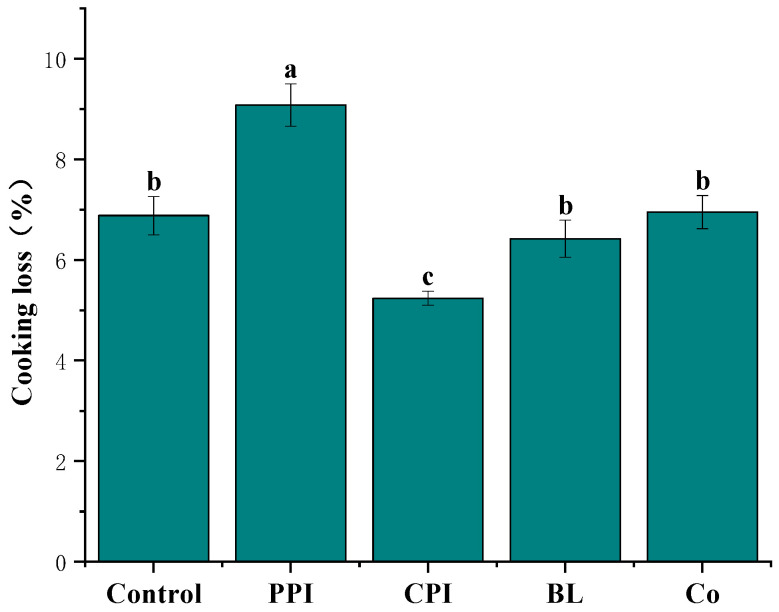
Effects of different protein pre-emulsified soybean oils on the cooking loss of fish sausage. Note: Different letters indicate that there is a significant difference between samples (*p* < 0.05). The error bars represent the standard deviation (*n* = 3).

**Figure 3 foods-11-03192-f003:**
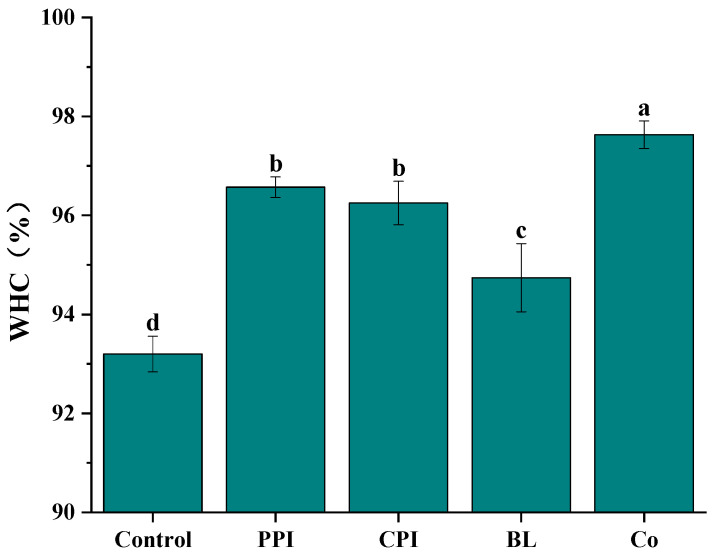
Effect of soybean oil with different protein pre-emulsification on the water retention of fish sausage. Note: Different letters indicate a significant difference between samples (*p* < 0.05). The error bars represent the standard deviation (*n* = 3).

**Figure 4 foods-11-03192-f004:**
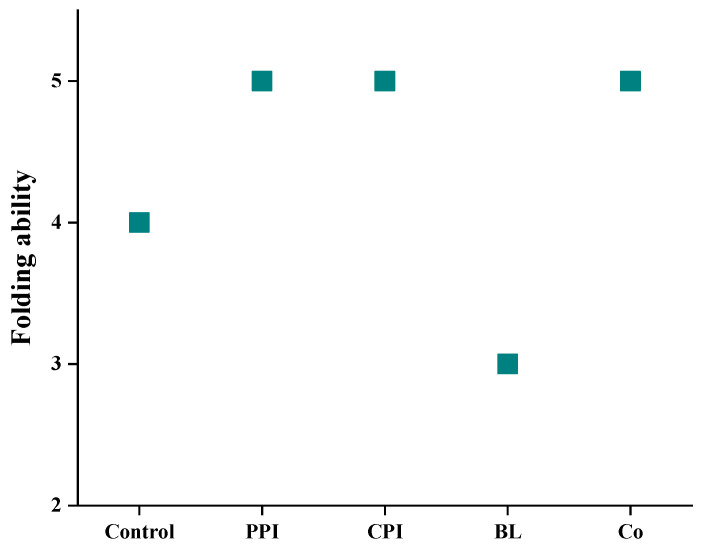
Effects of different protein pre-emulsified soybean oil on the folding capacity of fish sausage.

**Figure 5 foods-11-03192-f005:**
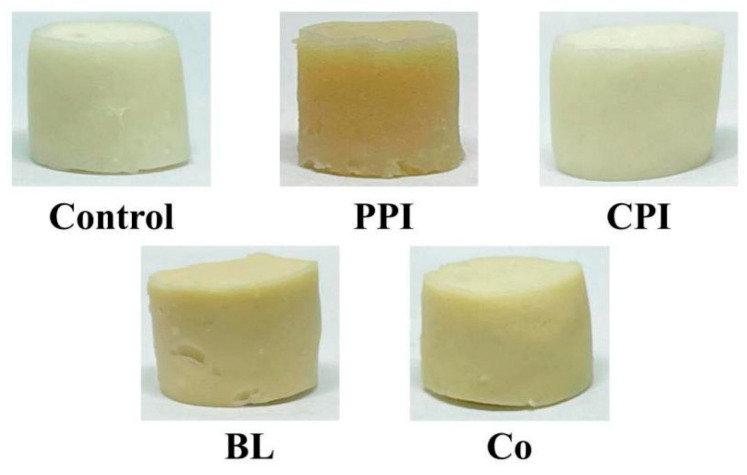
Appearance of soybean oil fish sausage pre-emulsified with different proteins.

**Figure 6 foods-11-03192-f006:**
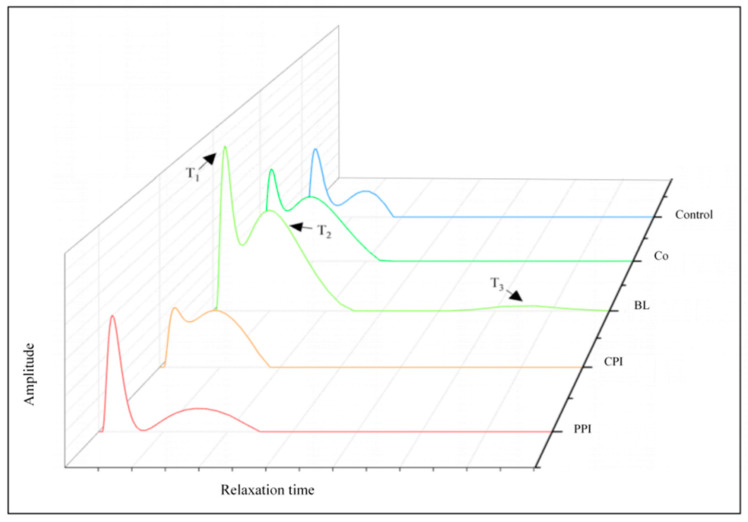
Effects of different protein pre-emulsified soybean oil on water distribution of fish sausage.

**Figure 7 foods-11-03192-f007:**
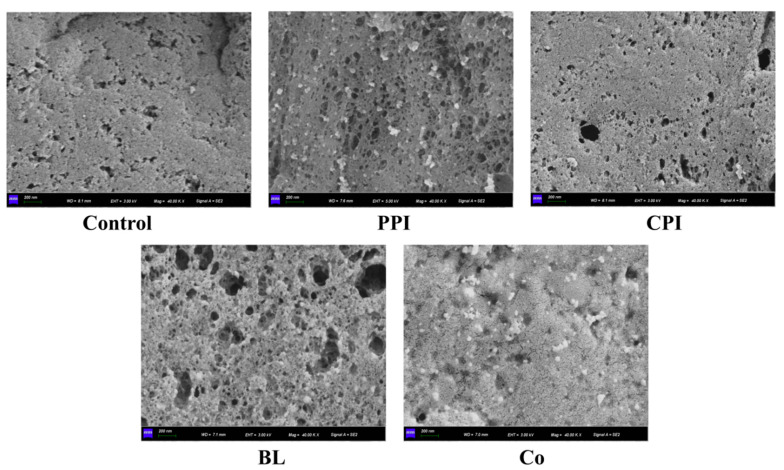
Effects of different protein pre-emulsified soybean oil on the microstructure of fish sausage (×40,000).

**Figure 8 foods-11-03192-f008:**
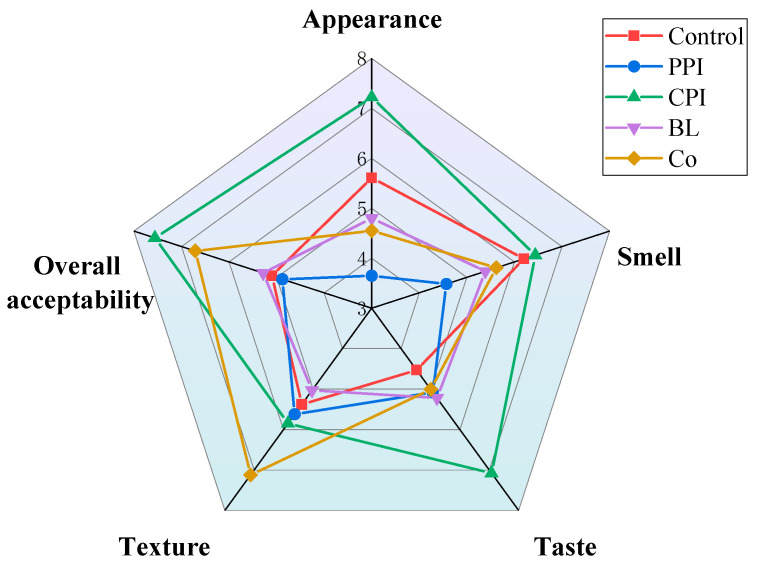
Sensory intensity radar diagram of fish sausage made with soybean oil and various pre-emulsified proteins.

**Table 1 foods-11-03192-t001:** Scoring criteria for sensory evaluation of grass carp sausage.

Project	Features	Score
Appearance (10)	Dark or grey, and there are numerous holes in the section	0.0–3.9
	Grey-white, with a few holes in the section	4.0–6.9
	Light yellow or fish white, bright colour; a few holes in section in the cross-section	7.0–10.0
Smell (10)	Fishy or bean fishy, unpleasant	0.0–3.9
	Light aroma or a slight characteristic fishy smell, without peculiar smell	4.0–6.9
	Ideal fish sausage has a strong aroma and a pleasant smell	7.0–10.0
Taste (10)	Oil tastes heavier or beans bitter taste heavier, difficult to swallow	0.0–3.9
	Mild astringency, umami, and saltiness are mild or heavy and can be tasted	4.0–6.9
	It has the taste of fish sausage, umami, and saltiness, worth tasting	7.0–10.0
Texture (10)	Too hard or too soft, too dry or too much water or oil, difficult to chew	0.0–3.9
	Normal, slightly uncomfortable to chew, but acceptable	4.0–6.9
	Good taste and suitable hardness, toughness, and elasticity	7.0–10.0
Overall acceptability (10)	Hard to accept ~ Good to accept	0.0–10.0

**Table 2 foods-11-03192-t002:** Relative band optical density (%) of BL and Co (non-reducing electrophoretogram).

Samples	Aggregates	MHC	Convicilin	Leg α + β	Vicilin	AC	Leg α	TM	Leg β	Others	Vicilin/Leg α + β
BL	11.61	15.90	4.60	9.98	7.22	19.31	10.63	5.31	9.98	5.45	0.72
Co	19.62	15.27	0.11	6.55	13.36	16.46	8.06	5.36	4.63	10.57	2.04

**Table 3 foods-11-03192-t003:** Chemical composition of PPI, CPI, Co, and BL.

Sample	Moisture	Protein	Fat	Carbohydrate	Ash
PPI	6.71 ± 0.14 ^a^	84.84 ± 0.03 ^c^	0.35 ± 0.02 ^d^	5.89 ± 0.19 ^a^	2.21 ± 0.16 ^a^
CPI	5.30 ± 0.07 ^b^	85.30 ± 0.13 ^b^	6.15 ± 0.05 ^a^	1.68 ± 0.13 ^c^	1.57 ± 0.11 ^c^
BL	4.01 ± 0.05 ^c^	85.07 ± 0.09 ^b^	5.24 ± 0.03 ^b^	3.79 ± 0.04 ^b^	1.89 ± 0.09 ^b^
Co	5.28 ± 0.11 ^b^	87.91 ± 0.02 ^a^	4.84 ± 0.08 ^c^	0.16 ± 0.08 ^d^	1.81 ± 0.01 ^b^

Note: Different letters in the same column indicate a significant difference between samples (*p* < 0.05).

**Table 4 foods-11-03192-t004:** Effects of different protein pre-emulsified soybean oil on the fish-sausage texture.

Sample	Hardness (g)	Springiness (mm)	Cohesiveness	Chewiness (N·mm)
Control	354.581 ± 5.252 ^e^	0.885 ± 0.003 ^a^	0.575 ± 0.004 ^c^	170.868 ± 4.234 ^d^
PPI	562.330 ± 9.359 ^c^	0.812 ± 0.002 ^e^	0.597 ± 0.009 ^b^	272.554 ± 7.860 ^b^
CPI	586.078 ± 13.727 ^b^	0.832 ± 0.003 ^d^	0.580 ± 0.008 ^c^	282.826 ± 7.474 ^b^
BL	424.238 ± 7.348 ^d^	0.867 ± 0.006 ^b^	0.554 ± 0.005 ^d^	207.450 ± 6.173 ^c^
Co	659.296 ± 12.823 ^a^	0.854 ± 0.005 ^c^	0.622 ± 0.006 ^a^	350.302 ± 9.637 ^a^

Note: Different letters in the same column indicate a significant difference between samples (*p* < 0.05).

**Table 5 foods-11-03192-t005:** Effect of soybean oil with different protein pre-emulsification on the colour of fish sausage.

Sample	L*	a*	b*	Whiteness
Control	83.51 ± 0.03 ^b^	−1.42 ± 0.10 ^c^	17.93 ± 0.22 ^d^	75.60 ± 0.15 ^b^
PPI	76.42 ± 0.11 ^e^	0.64 ± 0.05 ^a^	23.69 ± 0.03 ^a^	66.57 ± 0.07 ^e^
CPI	86.37 ± 0.10 ^a^	−2.37 ± 0.01 ^d^	8.89 ± 0.10 ^e^	83.56 ± 0.03 ^a^
BL	82.37 ± 0.15 ^c^	−1.14 ± 0.05 ^b^	18.31 ± 0.03 ^c^	74.56 ± 0.08 ^c^
Co	80.75 ± 0.04 ^d^	−1.15 ± 0.03 ^b^	19.37 ± 0.03 ^b^	72.67 ± 0.01 ^d^

Note: Different letters in the same column indicate a significant difference between samples (*p* < 0.05).

## Data Availability

Research data are not shared.

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
