# Peer review of "Effect of Pre-Emulsion of Pea-Grass Carp Co-Precipitation Dual Protein on the Gel Quality of Fish Sausage"

_foods, 2022, doi:10.3390/foods11203192_

Round 1
Reviewer 2 Report
General comment
This manuscript deals with the use of pea protein isolate along with grass carp protein via precipitation process, which was further used to pre-emulsify the vegetable one for sausage production. It is kind of interesting to use plant-based protein as the part of gel, especially to enhance water holding capacity of gel. However, the gel strength slightly decreased.
Specific comments
1. Please provide the details for preparation of BL and Co. It is hard to follow at the present form, especially for Co preparation. All the important conditions must be given in detail.
2. When the PPI and CPI were mixed at 1:1 ratio. W/W was stated. Please clarify if it is dry weight or not. It must be dry weight since the moisture content of both proteins could be different as affected by extraction process.
3. What was the protein patterns of all the four samples? This data is important in term of protein point of view.
4. Why the ratio of 12: 44: 44 was used to prepare the pre-emulsion. Please explain in the text. Was this mixture homogenized? Authors mentioned only stirring. How did the mixture become homogenous? Please clarify along with giving more details.
5. Provide the diameter and length of sausage casing.
6. For cooking loss, how could authors ensure that the water drip caused by cooking was remove completely? Please give the details.
7. Folding test has not been used for many years. Please remove figure and text from the manuscript. It is somehow obsolete.
8. The color was affected by sodium nitrite. What was the general color of sausage? It seems like it did not increase redness (no change in a* value). Please discuss in the text for the purpose of addition of sodium nitrite.
9. For SEM, why did authors remove the oil? Oil was supposed to impart the textural and microstructure of sausage? Thus, it is supposed to be localized in the sausage. Please elaborate and add the reason in the methodology. In fact, oil must be fixed in the sample.
10. For sensory evaluation, it is not correct. Please amend the methodology. First, authors used only 10 panelists. Were they experts? If so, the data must be changed to intensity for each attribute, NOT acceptability.
11. For overall acceptability, only score must be given by panelists without any guidance. That is because preference or acceptability was depending on individual and was not be guided. Authors can report only the values for overall acceptability in the text.
12. Bar in all figures must be defined, e.g. ‘Bars represent the standard deviation (n=3).
13. Table 2, please provide other texture attributes, such as cohesiveness, adhesiveness, etc. The discussion must be extended. For the control, the proteins were excluded. As a result, the amount of protein in the sample was lower, causing the poorer gel textural properties. Authors must discuss this point in the text.
14. Table 2 and 3. Please check the footnote. It should be changed to ‘Different letters in the same column indicate………………..’
15. Figure 6 caption: The magnification must be stated.
16. Figure 7. Please redraw by removal of overall acceptability. Also, use ‘Intensity’ in state of ‘Sensory score’ in the figure caption.
17. Protein patterns (SDS-PAGE) of different gels are required for this study. Cross-linking or degradation of proteins can be elucidated. Those changes could determine the sausage properties.
Round 2
Reviewer 1 Report
Thank you for the revision. Few minor comments:
- Line 96-98: It is not a sufficient scientific motivation for a paper that the research group has not studied something yet.
- The aim is now included in the introduction, but contains the word "better". As already made clear in my first review, this is unscientific language, please correct throughout the manuscript.
- After the aim in the introduction, there are still hypotheses missing.
- What is the purpose of Figure 1? It is very repetitive and takes up a lot of space.
- There are still spelling and grammar mistakes throughout the manuscript (e.g., line 106 "provide").
Reviewer 2 Report
Dear Editor,
After revision, some important details and discussion have been provided. The text has more clarity. However, some points must be addressed and the important data must be provided.
1. For the sensory evaluation, the term used for likeness or acceptability was not correct. Please change from '0 (dislike extremely' to '0 (poor sensorial property' and change from '10 (like extremely' to '10 Excellent sensorial property'.
2. Table 8 regarding overall likeness must be removed. It was repeated data with those appearing in Figure 8. However, the wrong number of Table and Figure is given and authors need to check and correct.
3. The data on protein pattern must be included in the text. Authors stated that proteins are essential part of this study. Thus, protein patterns should be provided and some discussion must be made.
